# Siamese Tracking of Cell Behaviour Patterns

**Andreas Panteli**∗                                                   ANDREAS.PANTELI@STUDENT.UVA.NL
**Deepak K. Gupta**∗                                                              D.K.GUPTA@UVA.NL
**Nathan de Bruijn**                                                       NATHANLDEBRUIJN@GMAIL.COM
**Efstratios Gavves**                                                            E.GAVVES@UVA.NL
*Informatics Institute, University of Amsterdam, The Netherlands*

## Abstract

Tracking and segmentation of biological cells in video sequences is a challenging problem, especially due to the similarity of the cells and high levels of inherent noise. Most machine learning-based approaches lack robustness and suffer from sensitivity to less prominent events such as mitosis, apoptosis and cell collisions. Due to the large variance in medical image characteristics, most approaches are dataset-specific and do not generalise well on other datasets.

In this paper, we propose a simple end-to-end cascade neural architecture that can effectively model the movement behaviour of biological cells and predict collision and mitosis events. Our approach uses U-Net for an initial segmentation which is then improved through processing by a siamese tracker capable of matching each cell along the temporal axis. By facilitating the re-segmentation of collided and mitotic cells, our method demonstrates its capability to handle volatile trajectories and unpredictable cell locations while being invariant to cell morphology. We demonstrate that our tracking approach achieves state-of-the-art results on PhC-C2DL-PSC and Fluo-N2DH-SIM+ datasets and ranks second on the DIC-C2DH-HeLa dataset of the cell tracking challenge benchmarks.

**Keywords:** Cell tracking, re-segmentation, mitosis, cell collision, Siamese tracker

## 1. Introduction

Advancements in the field of machine learning (ML) have allowed for the automation of medical data analysis, with several advanced methods performing as good as humans on a range of tasks (*e.g.*, Dalca et al. 2018; Chen et al. 2018). Example problems where deep learning (DL) methods have produce significant impact include detection and classification of tumours (Coudray et al., 2018; Kamnitsas et al., 2017; Budinska et al., 2013; Kalaiselvi and Nasira, 2014), identification of new biomarkers in high-dimensional data (Budinska et al., 2013; Ravi and Hegadi, 2015; Rathi et al., 2011), among others.

The problem of medical data analysis becomes more challenging at the micrometer scale (*e.g.* pancreatic stem cells or cell nuclei), where it is difficult to visualise and process the data (Coudray et al., 2018; Saltz et al., 2018). Cell distinctive shapes and morphological traits are convoluted with low resolution artefacts, noise components from the microscope scanning device and varying lighting conditions (Swiderska-Chadaj et al., 2019). Figure 1 shows example images of cells for five different datasets from the IEEE International Symposium on Biomedical Imaging (ISBI) cell tracking challenge (Maška et al., 2014). As

---

∗ Contributed equally

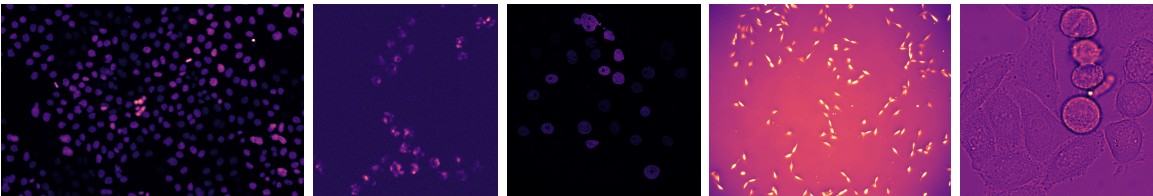

Figure 1: Example images of (a) Fluo-N2DL-HeLa (b) Fluo-N2DH-SIM+ (c) Fluo-N2DH-GOWT1 (d) PhC-C2DL-PSC (e) DIC-C2DH-HeLa datasets from the cell tracking challenge (Maška et al., 2014).

can be seen, there are various challenges such as low signal-to-noise ratio, poor illumination, clutter of cells and occlusion that make it difficult to accurately track and segment the cells with a naked eye.

In addition to the challenges outlined in Figure 1, biological cells also fail to conform to a predefined shape and there can be several shifting movement patterns that are are hard to analyse or detect (Saltz et al., 2018). Examples of such behaviour are shown in Figure 4 where a parent cell gets split into two daughter cells (Mitosis) or two cells collide and appear as a single cell (Collision).

Some cell segmentation and tracking approaches include constructing temporal trajectories for cells (Yang et al., 2005), spatial correlation using Delaunay graphs (Nath et al., 2006), and watershed deconvolution with morphological operators (Sharif et al., 2012). Due to the large noise components of medical images, ML methods mediating dataset-specific properties such as uneven illumination and lack of pixel-value normalisation, tend to define many morphological dependent conditions for every cell type (Sharif et al., 2012; Lux and Matula, 2019). However, this process is sensitive to outliers and along with volatile positional changes, it makes tracking each individual cell over time more difficult. This is because a cell in the previous frame might change completely in the next.

The U-Net neural network (NN) introduced by Ronneberger et al. (2015) is among the most successful NN architectures in biomedical image segmentation for tasks such as tumour detection and living cell segmentation (Heller et al., 2019; Falk et al., 2019). Due to its U-shape and essential residual connections, U-Net has achieved state-of-the-art results in many challenges (Li et al., 2018; Dubost et al., 2017). However, despite its high performance, even U-Net struggles on images containing significant movements of cells and changes in their morphology, and in particular, fails to reliably detect cells that split or die (leave the field of view of the scanning device) (Christ et al., 2016). Lux and Matula (2019) combined U-Net with watershed deconvolution (Kachouie et al., 2008) and demonstrated improved performance on cell tracking datasets[1]. However, this approach involves tuning of several parameters, *i.e.*, several data-specific intermediate processing steps relating to cell size, erosion, staining, and other morphological traits so as to acquire reasonable cell shapes. Zhou et al. (2019) proposed using two U-Net architectures, one for segmenting cells and the other for detecting their centroids. Due to learning data specific network weights for cell detection, this approach is more resilient to outliers, however, it does not suggest a robust way to detect smaller cells that are falsely segmented as a single cell.

---

1. *http://celltrackingchallenge.net/*

Lux and Matula (2019) used static area overlap for building the correspondence of cell trajectory in subsequent frames. Another approach uses level sets to follow the evolution of cells (Maška et al., 2014). However, due to the fluidic and erratic nature of cells, neither of these algorithms are able to model their real movement patterns. Siamese networks, first introduced in the work of Bromley et al. (1994) and adapted for Siamese Instance Search Tracking (SINT) by Tao et al. (2016), have shown to excel in generic object tracking (Bertinetto et al., 2016). Due to their robustness in object matching under appearance variations, siamese methods have been useful in segmenting medical images (Spitzer et al., 2018).

In this paper, we introduce a re-segmentation approach, which relies on siamese matching-based trackers, combined with U-Net and the watershed deconvolution method, to track cells and model cell collisions, mitosis and apoptosis. Compared to the original siamese trackers, as in the work of Tao et al. (2016) and Bertinetto et al. (2016) proposed on natural image videos, we model cell behavioural patterns in order to correctly track cells. Preliminary results related to this research were recently reported by us in (Gupta et al., 2019). Compared to the previous cell tracking methods, such as in the work of Magnusson (2016) and Lux and Matula (2019), we generalise tracking to an approach independent of cell-type, predicting cell displacement across frames.

Our proposed Tracking-Assisted Segmentation (TAS) can easily model and detect the unstable biological cell movement activities such as mitosis and cell collisions that lead to false predictions. Further, through the use of the watershed algorithm, false predictions can be re-segmented to improve the overall cell tracking performance. The contributions of this paper can be summarised as follows.

- We augment segmentation by siamese tracking for improved temporal correspondence and re-segmentation of erroneous predictions.

- Our approach is more robust to morphology variations and explicitly models rare events such as mitosis, apoptosis and cell collisions.

- Our approach generalises well to different datasets outperforming published state-of-the-art segmentation methods for biological cells on three benchmark datasets [2].

Our method can improve the autonomous segmentation of biological cells with the goal of inspecting multiple patient images in parallel, speed up diagnosis and ease the workload of doctors. Our approach indicates a more robust method of cell segmentation which will reduce the number of cells incorrectly detected and improve the accuracy and performance of such automatic systems.

## 2. Methodology

The method proposed here is a two step process with input being the raw images and the final output being per pixel classification labels as shown in Figure 3. The pipeline of our approach is shown in Algorithm 1.

---

2. For the DIC-C2DH-HeLa dataset, we achieved second place at the online ISBI cell tracking competition but the published approach of the team at first place, (Lux and Matula, 2019), is still outperformed by our method.

---

**Algorithm 1:** Tracking-Assisted Segmentation (TAS)

---

**Input:** Sequence of frames

**Output:** Per frame cell segmentation and tracking information

**for** *frame in sequence* **do**

> **Step 1**: Initial segmentation
>
> - Separate cell bodies from the background using U-Net
> - Define cell identities using watershed or random walker
>
> **Step 2**: Siamese tracking
>
> - Track each cell in small intervals
> - Identify biological cell behaviours:
>
>   - **Collision**, if $match(\text{cell}_t, \text{cell}_{t-1}^{(1)}, \text{cell}_{t-1}^{(2)})$
>   - **Mitosis**, if $match(\text{cell}_t, \text{cell}_{t+1}^{(1)}, \text{cell}_{t+1}^{(2)})$
>   - **Cell death**, if $match(\text{cell}_t, \text{None})$
>
> - Re-segment cells using watershed

**end**

---

### 2.1. Initial Segmentation

For the initial segmentation of cells, we propose three variants of our tracking-assisted segmentation (TAS) method, namely general, intermediate and specialised TAS. Here and henceforth, these approaches will be referred as g-TAS, i-TAS and s-TAS, respectively. Further, the term *pre-processing* refers to thresholding and filtering the raw image, e.g., to account for varying illumination, cell size, morphology and so on. With *post-processing*, we refer to additional adjustments after the U-Net output, such as filling prediction gaps, threshold smaller regions and fitting ellipses. We follow Lux and Matula (2019) for the pre- and post-processing steps, either per dataset -as originally in their work- or across datasets. The three TAS variants differ in the pre- and post-processing steps and how these steps involve parameter tuning to a specific dataset.

1. g-TAS performs no pre-processing and no post-processing that relates to the U-Net approach or any specific cell-type.

2. i-TAS tunes the pre-processing hyperparameters (associated with image normalisation, histogram equalisation and uneven illumination correction) independently per dataset (on the respective training sets) as explained in the work of Lux and Matula (2019). This is to account for the very different range and distribution of pixel values that adversely affect the U-Net predictions. The siamese-based re-segmentation step after the U-Net output (our main contribution), remains the same for all data sets.

3. s-TAS adapts the pre- and post-processing hyperparameters per dataset, using the approach proposed in Lux and Matula (2019). This variant is designed specifically

for detecting precise cell boundaries in order to minimise erroneous cell detection and be competitive with optimised state-of-the-art approaches. s-TAS retains the same re-segmentation augmented approach which we introduce.

### 2.1.1. General Tracking Assisted Segmentation (g-TAS)

G-TAS is a generalised implementation of tracking-assisted segmentation approach that requires no tuning or adjustments with respect to any datasets. First, a U-Net model (Ronneberger et al., 2015) is explicitly trained across all data from all the different cell types, 3.1. Next, the random walker algorithm from the work of Grady (2006) is used to generate the initial cell segmentation of the pipeline. Recent methods use the watershed algorithm for splitting cell body predictions from neural networks into smaller cells (Sharif et al., 2012; Lux and Matula, 2019). However, the random walker algorithm is shown to be more resilient to noise and hence more suitable for predictions of diverse image properties and cell morphologies (Grady, 2006). The output of the random walker is then fed into the tracker network to correct for mitosis, apoptosis and cell collisions, 2.2.2.

Training of the U-Net network was done using data augmentation with additive noise, pixel value range shift and a cutoff on the maximum value. The choice of these specific augmentations arises from the observation of the dataset images which are not consistent in a standard RGB value range and have low signal-to-noise ratio as indicated by Maška et al. (2014). The network is trained with a learning rate of 0.001 using the Adam optimiser for 50 epochs. Normalisation of the raw images includes (i) normalisation to zero mean and unit variance, (ii) construction of cell centroids from the U-Net predictions based on the distance to the background (where 0 indicates background and 1 the farthest foreground), and (iii) distinct cells are defined by running the random walker algorithm. All parameters in g-TAS are the same across all datasets. More details related to the experimental implementation are presented in Appendix A. Unlike the methods that use thresholds based on morphology (Lux and Matula, 2019), the pre-processing of the raw images and post-segmentation division of small cells are also generalised, and not specific to any dataset.

### 2.1.2. Intermediate Tracking Assisted Segmentation (i-TAS)

This approach relaxes the constraint of a general pre-processing method and uses the U-Net implementation of Ronneberger et al. (2015), pre-trained as in the work of Lux and Matula (2019). The same pre-processing steps are employed, and the model is manually fine-tuned to the image properties based on the type of cell image. To evaluate the effect of morphological boundary refinements on cell segmentation, the approach of defining unique cells from the initial segmentation, using the random walker algorithm, is kept the same as in g-TAS.

### 2.1.3. Specialised Tracking Assisted Segmentation (s-TAS)

The third TAS variant relaxes further the constraint of dataset-tuned segmentation, training the U-Net in each dataset separately, and applies the water deconvolution method as described in the work of Lux and Matula (2019) instead of the random walker algorithm. Compared to i-TAS, the difference is that the post-processing implementation of our model also now uses several tuning parameters which are typically adjusted for every dataset.

Thus, this approach can be interpreted as a fine-tuned implementation of TAS that specialises for a certain dataset.

The output of the initial segmentation (g-TAS, i-TAS or s-TAS without the siamese re-segmentation) produces finely segmented cells where the actual cell behaviours might not be correctly expressed. A schematic representation related to this issue is shown in Figure 2. To be able to correctly reason about cell movement over time, cells need to be correctly tracked as well.

## 2.2. Siamese tracking

The location of the cell in subsequent frames is linked using siamese tracking. For tracking purposes, a SiamFC tracker (Bertinetto et al., 2016), pre-trained on the GOT-10k dataset (Huang et al., 2019), is used. Tracking is done in the forward as well as in the backward direction to predict the new location of any cell in the previous and next frames based on its position in the current frame. The cell segmentation is refined through tracking by detecting the occurrences of mitosis and collision events. For the two events, tracking is performed in opposite directions along the temporal dimension.

The working of the tracking module is as follows. Let $I_t$ denote the $t^{\text{th}}$ frame in a sequence of length $T$, and $\mathcal{S}_t = \{C_t^1, ..., C_t^K\}$ be the set of detected cells in this frame. These are used to initialise the tracker at step $t$. For cell $C_t^i$, the predicted locations by the tracker in $I_{t+1}$ and $I_{t-1}$ are referred as forward $(F_t^i)$ and backward $(B_t^i)$ predictions, respectively. Collision and mitosis are then detected, descriptions of which follow below. Note that the movement prediction model explained here does not depend on the morphology of the cell or on any other image property. Hence, it is directly applicable on top of any segmentation algorithm without the need for additional tuning.

It is important to note that the tracker uses a step size of 1, either in the positive or negative temporal dimension. We have experimented with other step sizes but found that mitotic events are not consistent in step size, and from our parameter search, step size of 1 seemed appropriate. However, cells may collide with each other and often stay close to other cells, having adjacent boundaries for several frames, before they split up again. When cells coincide with one another for more than one frame then the re-segmentation will split them at the first possible frame where it detects a collision or mitosis. Thereafter, from that frame on wards, the algorithm will track both of them; re-segmenting them again if they are collided in subsequent frames. In the event of a false negative, e.g. a mitotic event that is not detected, the algorithm will assume the cell to be one bigger cell. In the next frame, if the mitotic event becomes more clear, i.e. the algorithm detects two independent cells, then the algorithm will re-segment the bigger cells into two, one frame later.

### 2.2.1. Collision detection

Collision occurs when two cells share a fraction of their boundary, and this can be mistaken as a single cell during segmentation. When processing a new frame $I_t$, where $t > 1$, collision detection is performed first in which a cell $C_t^i$ is considered to be a lump of multiple individual cells if the centroids of two or more cells in $\mathcal{S}_{t-1}$ lie within the tracked region $B_t^i$. If this is the case, $C_t^i$ is re-segmented using the centroids of the two cells in the previous frame

$I_{t-1}$. This collision detection procedure continues until each cell in $I_t$ matches at most one cell in $I_{t-1}$.

### 2.2.2. MITOSIS DETECTION

For detecting mitotic events, a procedure similar to that of detecting collisions is performed on a sequence of frames, but in the reverse direction. Cells are matched in $\mathcal{S}_{t-1}$ to the detected cells in $I_t$. Namely a cell $C^i_{t-1}$ is matched to a cell $C^i_t$ if the centroid of $C^i_t$ is inside the region $F^i_{t-1}$. Different from collision detection, however, $C^i_{t-1}$ is also matched to $C^i_t$ if the centroid of the region $F^i_{t-1}$ lies within the boundaries of the cell $C^i_t$. This matching procedure yields a set of matches for each cell $C^i_{t-1}$, which we denote as $M^i_{t-1}$ and its size as $|M^i_{t-1}|$. $|M^i_{t-1}|$ indicates the number of cells that are associated (matched) in the subsequent frame, with cell $C^t_{t-1}$ in the previous frame. If the number of matched cells is zero ($M^i_{t-1} = 0$) then the cell has died; if the number of matched cells is one ($M^i_{t-1} = 1$) then there is only one cell in the next frame associated with the cell in the previous frame; if $M^i_{t-1} > 1$, then there exists a mitotic event. The state of cell $C^i_{t-1}$ is then determined according to Equation (1).

$$C^i_{t-1} = \begin{cases} \text{Apoptosis,} & |M^i_{t-1}| = 0 \\ M^i_{t-1,1}, & |M^i_{t-1}| = 1 \\ \text{Mitosis,} & \text{otherwise} \end{cases} \tag{1}$$

where apoptosis is the cell death, or the disappearance of the cell from the field of view, as described by Ulman et al. (2017).

In case of mitosis, the cell splits, thus the tracking of $C^i_{t-1}$ ends and the cells in $|M^i_{t-1}|$ are initialised with two new trackers which have $C^i_{t-1}$ as their parent. The new cells in $\mathcal{S}_t$ that are not linked to any cell in $I_{t-1}$ are interpreted as newly detected cells which start their "life" in $I_t$ without link to a parent cell.

### 2.2.3. RE-SEGMENTATION

In case of a detected collision of two or more cells into a cell $C^i_t$, cell $C^i_t$ is re-segmented in such a manner that the new number of segments matches the number of colliding cells. This is achieved using watershed deconvolution as described in the work of Kachouie et al. (2008). To prevent over-segmentation of the cell $C^i_t$, which adversely affects segmentation accuracy, the relative position of the centroids of the cells in $I_{t-1}$ are used as the seeds for the segmentation algorithm. An illustration showing re-segmentation of cells is shown in Figure 2.

The above algorithm is designed as a simple and general method to track biological cells of different size, shape and movement patterns. Volatile trajectories and unpredictability of cell location are dealt with using one approach for all datasets, with the re-segmentation being invariant to cell morphology and image properties.

## 3. Experiments

### 3.1. Data

The effectiveness of the proposed methodology is demonstrated through applications on three datasets listed on the ISBI cell tracking challenge[1] (Ulman et al., 2017; Maška et al., 2014), namely PhC-C2DL-PSC, Fluo-N2DH-SIM+, and DIC-C2DH-HeLa. They contain stem cells and nuclei of cells, respectively, shaped as small ellipsoids, as shown in Figure 1. The PhC-C2DL-PSC dataset contains significantly larger number of cells and an active population of cells moves very erratically (Maška et al., 2014). For Fluo-N2DH-SIM+, the cell movements are moderate, and for DIC-C2DH-HeLa, they are minimum. For DIC-C2DH-HeLa, the cell morphology deviates from a round circle and lacks a well defined perimeter boundary due to small cell structures on the edge of the cells. This characteristic makes the cell segmentation more difficult when combined with the other datasets to be segmented by one single approach. Two sequences per dataset are available for training, and another set of two sequences per dataset is used for testing.

### 3.2. Results

The performance of our methods is evaluated using the same measures as described in Maška et al. (2014). For segmentation, the average of detection (DET) and segmentation (SEG) metrics, expressed as $OP_{CSB} = \frac{1}{2}(DET+SEG)$, is used. Further, for tracking, the average of SEG and tracking (TRA) measures, expressed as $OP_{CTB} = \frac{1}{2}(SEG+TRA)$, is used.

Table 1 provides the performance scores for the three variants of TAS. As expected, the performance of s-TAS is the best among the three proposed variants, while that of g-TAS is the lowest. Comparing the published works of Zhou et al. (2019) and Lux and Matula (2019) with the TAS variants, on the Fluo-N2DH-SIM+ dataset, shows that our g-TAS approach outperforms one of the baselines set by Zhou et al. (2019). The intermediate approach, i-TAS, further improves the performance, primarily due to the more precise boundary refining of ellipsoid shapes. For the PhC-C2DL-PSC and DIC-C2DH-HeLa datasets, g-TAS exhibits a significant drop in performance. This is due to overfitting mainly because of lower number of training samples compared to the Fluo-N2DH-SIM set; when all data is used to train one model. In the i-TAS and s-TAS variants however, the U-Net is trained on each dataset separately which improves performance significantly and the s-TAS variant outperforms both baselines.

We further compare the performance of our approach with the top 3 submissions, state-of-art methods, on the leaderboard of ISBI cell tracking challenge[1]. Approaches in this challenge are precisely tuned to the type of cell morphology and image properties in order to achieve competitive performance. Table 2 presents the results of our method compared to the other approaches. The detailed results, including all metrics, are shown in Table 4 in Appendix C. As can be seen, our approach outperforms all other methods for datasets PhC-C2DL-PSC and Fluo-N2DH-SIM+, and ranks second for the DIC-C2DH-HeLa.

An interesting observation is that for the tracking metric (TRA) for the DIC-C2DH-HeLa dataset, 4, our approach outperforms the rest with a value of 0.955. The cells in DIC-C2DH-HeLa lack easily discernible boundaries compared to other datasets and also

|  | DIC-C2DH-HeLa | | Fluo-N2DH-SIM+ | | PhC-C2DL-PSC | |
|---|---|---|---|---|---|---|
| Method | $OP_{CSB}$ | $OP_{CTB}$ | $OP_{CSB}$ | $OP_{CTB}$ | $OP_{CSB}$ | $OP_{CTB}$ |
| Zhou et al. (2019) | - | - | 0.861 | 0.860 | 0.806 | 0.801 |
| Lux and Matula (2019) | 0.894 | - | - | - | - | - |
| g-TAS | 0.077 | 0.076 | 0.872 | 0.870 | 0.315 | 0.308 |
| i-TAS | 0.880 | 0.874 | 0.895 | 0.893 | 0.717 | 0.704 |
| s-TAS | **0.905** | **0.904** | **0.897** | **0.896** | **0.846** | **0.843** |

Table 1: $OP_{CSB}$ and $OP_{CTB}$ results of the three different initial segmentation approaches, as described in section 2 for the published state-of-the-art approaches.

|  | DIC-C2DH-HeLa | | Fluo-N2DH-SIM+ | | PhC-C2DL-PSC | |
|---|---|---|---|---|---|---|
| Method | $OP_{CSB}$ | $OP_{CTB}$ | $OP_{CSB}$ | $OP_{CTB}$ | $OP_{CSB}$ | $OP_{CTB}$ |
| ISBI CTC[1] 3rd entry | 0.884 | 0.848 | 0.887 | 0.882 | 0.808 | 0.804 |
| ISBI CTC[1] 2nd entry | 0.895 | 0.894 | 0.890 | 0.889 | 0.809 | 0.804 |
| ISBI CTC[1] 1st entry | **0.912** | **0.909** | 0.896 | 0.895 | 0.841 | 0.836 |
| s-TAS | 0.905 | 0.904 | **0.897** | **0.896** | **0.846** | **0.843** |

Table 2: $OP_{CSB}$ and $OP_{CTB}$ scores, as of 30th of January for the entries to the ISBI Cell Tracking Competition (CTC)[1] (Maška et al., 2014). MU-Lux-CZ the same team as in the work of Lux and Matula (2019), ND-US, BGU-IL the same team as in the work of Zhou et al. (2019), CVUT-CZ, HD-Hau-GE, UVA-NL (our earlier submission of this work).

lack intense movement activities. This is primarily the reason that this dataset does not benefit from our modelling of cell behaviour. The minor shift of the TRA metric can be explained by the explicit collision detection method which the method of Lux and Matula (2019) lacks. In datasets Fluo-N2DH-SIM+ and PhC-C2DL-PSC, highly active cells are better detected by our tracking method. This is clearly evident in PhC-C2DL-PSC, where up to 1000 cells are present in each frame, compared to a maximum of around 70 in the Fluo-N2DH-SIM+. Collisions and mitotic events are more evident with increased spatial displacement which constitutes our method an ideal tracker and re-segmentation. Due to this reason, our method outperforms the previous state-of-art-approaches by a larger margin on this dataset.

The code used to produce these results is available at
*https://gitlab.com/Baggsy/cell_tracking_2019*

## 4. Discussion

Since our approach enhances the tracking capabilities of segmentation focused methods, it is important to identify which aspects of the tracker contribute most to the improvements reported earlier. Table 3 presents an ablation study involving s-TAS being studied with respect to the collision and mitosis detection modules. As can be seen, both collision and mitosis modelling steps are an integral part of the approach, since there exists a clear dependence on them. Removing the re-segmentation correction completely results in lower performance values.

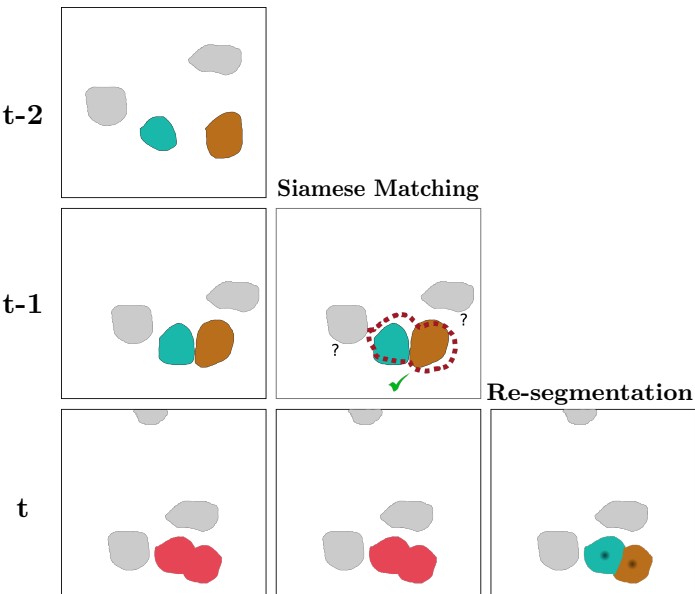

Figure 2: Schematic representation of cell collision detection using a Siamese tracker, and re-segmentation of the detected cells using watershed approach. The two cells, far apart in $(t-2)^{\text{th}}$ frame, collide in the $t^{\text{th}}$ frame, and are wrongly segmented as a single cell. Through siamese tracking between $(t-1)^{\text{th}}$ and $t^{\text{th}}$ frames, the collision event is identified, and applying the watershed approach over the $t^{\text{th}}$ frame, using the cell centroids of frame $(t-1)^{\text{th}}$, helps to correct the segmentation.

However, it is important to note that the performance variation between different model variants are not very large, with a mean standard deviation value of 0.0169. Upon further inspection of the 2nd and 3rd entries for all datasets in Table 4, all differences seem marginal, indicating saturation in the benchmark. We further explore this in our findings of noisy labels with these properties: (1) Delayed mitotic events (a single label for 2 distinct cell boundaries) (2) Pre-mitotic events (two labels for one bigger cell body before a mitosis) (3) Cell death artefacts (4) Irregular shape of ground truth labels (label not matching visual inspection of cell). To adjust for delayed and pre-mitotic events we experimented with the mitotic/collisional event timing by hastening and delaying the re-segmentation but noticed a decrease in performance. These noisy labels are mostly due to the staining procedure and hence it is something all participating algorithms face.

For high cell activity, such as in the PhC-C2DL-PSC dataset, we observe a bigger increase in performance due to our ability to correct the initial segmentation in difficult-to-observe cell collisions. Last, we emphasise that quantitatively speaking, we consider as much more important the fact that we produce state-of-the-art results on several different datasets and without any dataset-specific algorithms (at most, we only tune hyperparameters on the respective training sets for the initial segmentation alone); rather than the final numbers themselves.

We notice that we improve particularly well when boundaries are not clearly distinguishable. From Figure 4, the mitotic event in the DIC-C2DH-HeLa dataset has a very visible

| (Base) s-TAS | | Fluo-N2DH-SIM+ | |
|---|---|---|---|
| Collision detection | Mitosis detection | $OP_{CSB}$ | $OP_{CTB}$ |
| + | + | **0.902** | **0.901** |
| - | + | 0.899 | 0.899 |
| + | - | 0.875 | 0.874 |
| - | - | 0.859 | 0.854 |

Table 3: $OP_{CSB}$ and $OP_{CTB}$ ablation study results on the **training** set using the s-TAS variant as a base method for applying modification on.

effect. Hence a simple segmentation can detect the mitotic event. In contrast, the collision example in Figure 4 for the Fluo-N2DH-SIM+ dataset, does not always result in clearly defined boundaries. As Figure 2 indicates, when single segmentation cannot split collided cells, our re-segmentation checks the next and previous frames for meshed bodies and is able to split them. The real power of the method, and where we get the biggest numerical improvements, is when there exist high cell activity in frames, such as collisions, for smaller cells such as is the case for the PhC-C2DL-PSC dataset due to the convoluted appearance of the cell bodies. We want to emphasise here that these advantages are inherent to the approach, and the core algorithm does not need to change (at most only hyperparameters tuned depending on the TAS variant).

## 5. Conclusions

Medical images of biological cells, contain several noisy artefacts, convoluted cell boundaries and unpredictable cell movements, which often confuse cell segmentation methods. We propose a siamese tracking assisted re-segmentation approach which specifically models biological cell activities (mitosis, apoptosis and cell collisions) and enhances the overall cell segmentation. Our results indicate that biologically inspired tracking models of micrometer-scaled cells can better apprehend erratic behaviour. We demonstrate the applicability of our method on three cell tracking datasets. The specialised variant outperforms the previous state-of-the-art models, and ranks first on two and second on one of the three benchmark datasets.

## Acknowledgments

We thank Martin Maška for his constant help, time and invaluable assistance.

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

## Appendix A. Implementation details

### A.1. Initial segmentation

The step by step approach to construct the initial segmentation of *g-TAS* is as follows. Raw images are normalised first by equally distributing 256 bins of pixel values, using the cumulative distribution of its histogram (Kim, 1997; Pizer et al., 1987). To enhance the contrast between maximum bright spots, cell bodies, and background information in the image, the logarithm function is applied before the histogram equalisation procedure. This will relate areas closer to cell bodies and help determine more stable boundaries due to minimised effect of standalone maximum values. Next, the image values are projected to the range of 0-1, and then normalised further to zero mean and unit variance. This pre-processing step is applied the same way to all images of any dataset and fed into the U-Net network to produce cell body predictions.

Thereafter, the Euclidean distance from the background is used as a successive threshold metric. What this entails is acquiring regions where the distance is above 0.5, if 0 indicates background information and 1 is the farthest area from it. Thereafter, these regions are compared with the bigger regions cutoff at 0.1. If there exists a region that was not identified by the stricter cutoff value of 0.5, then the new region is considered to be a cell centroid. This is performed again for a cutoff value of 0, which coincides with the exact U-Net prediction. The last step is necessary to detect small cells appearing in the frame, which the cutoff values 0.1 and 0.5 would ignore. An additional condition for a cell centroid is that is at least bigger than the third of the mean cell size detected at each step. This condition is necessary to reject small outlier pixels. The final cell centroid regions are used as seeds for the random walker approach to define the boundary of each smaller cell contained in the initial cell body prediction of the U-Net.

### A.2. U-Net prediction (post-)processing for TAS variants

Using variants g-TAS and i-TAS the output of U-Net is processed the same for all datasets. First, centroids are constructed using the euclidean distance, of every pixel, to the background. Centroids are first defined by regions where the euclidean distance is above a threshold value of 0.5; where 0 indicates background and 1 the farthest foreground. This will produce small centroid regions and in some cases where cells are too small no centroids. To remedy this a threshold value of 0.1 is used to define bigger centroid regions which are combined with the smaller ones. If there exist overlapping centroids from the two sets, then smaller centroids are preferred to split up cells that are collided. This process is repeated another time for threshold value 0, to detect even the smallest cells exiting or entering the frame. To remove unwanted artefacts in the centroids set, regions below the third of the average centroid region size are removed. This results in distinct centroids for each cell in the frame. In order to define where the boundary of lightly collided cells is, the random walker algorithm is used, using as seeds the centroid regions calculated. The output of the random walker approach produces individual cells on a frame and the frame is then fed into the siamese tracker for re-segmentation of heavily convoluted cells and cell behaviour modelling. Using the s-TAS variant, the same (post-)processing is applied on the U-Net output as in the work of Lux and Matula (2019). Namely, cell centroids are constructed

using manually defined ellipsoid kernels, pixel value thresholds and filling gaps in the cell bodies. Every dataset is processed with a different set of parameter values tailored to its cell type. Then the watershed algorithm is run using the original U-Net prediction and the constructed centroids to produce individual cells in the frame. This frame is then fed to the siamese tracker for re-segmentation.

### A.3. Data augmentation

Based on Ulman et al. (2017) we add (1) random additive noise, (2) pixel value range shifts, and (3) maximum value cutoffs. Specifically, for each image we create 15 augmented images, 5 images per augmentation technique: (1) We add to the input image random Gaussian noise $N(0, std)$, with five possible $std$ standard deviations 0.1, 0.325, 0.55, 0.775, 1. (2) We add an offset to the input image pixel values, which is a random number drawn (per image) uniformly in the range of [-1, 1] . (3) We set the maximum or minimum value of an image to a cutoff value calculated as $cutoff = max * num$. $max$ is either the minimum or the maximum value of the image, chosen randomly. $num$ is a random number drawn from a uniform distribution in the range of [0, 1]. E.g.: An image with range [-3, 3] and $max = -3$ and $num = 0.5$, will result in an image of range [-1.5, 3] with any value smaller than -1.5 being set to -1.5.

### A.4. Computational complexity

Our end-to-end model, U-Net and SiamFC combined, comprises 25 convolutional layers. In comparison, Lux and Matula (2019) use a U-Net with 20 convolutional layers. And, Zhou et al. (2019) use two U-Net Networks for a total of 40 convolutional layers. Thus, our model sits between the two competitor methods. Generally, we needed around 5-10 minutes per sequence comprising 100 to 400 frames on a Geforce TitanX gpu.

## Appendix B. Additional Illustrations

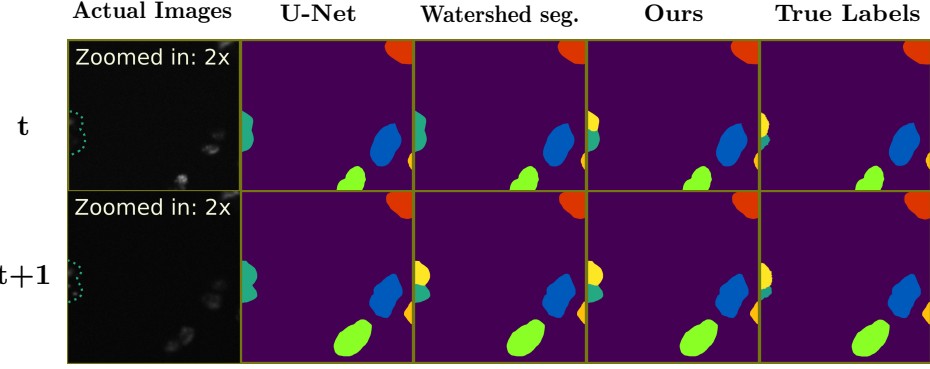

Figure 3: Cell segmentation illustration by (1) plain U-Net, (2) U-Net and watershed (3) Our $s$-$TAS$ method and (4) the true labels.

 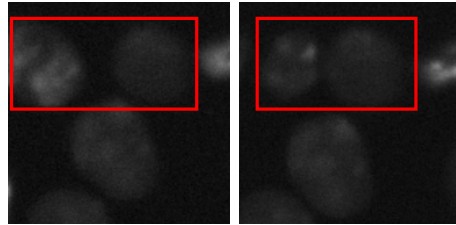

Mitotic event in DIC-C2DH-HeLa dataset      Collision event in Fluo-N2DH-SIM+ dataset

Figure 4: Biological cell movement behaviours between subsequent frames.



Figure 5: Example images of (a) Fluo-N2DL-HeLa (b) Fluo-N2DH-SIM+ (c) Fluo-N2DH-GOWT1 (d) PhC-C2DL-PSC (e) DIC-C2DH-HeLa datasets, as in Figure 1 in grayscale color map

## Appendix C. Additional Results

| Competition rank | DIC-C2DH-HeLa | | | | |
| --- | --- | --- | --- | --- | --- |
| | DET | SEG | TRA | OP$_{CSB}$ | OP$_{CTB}$ |
| 3rd entry | 0.948 | 0.820 | 0.909 | 0.884 | 0.848 |
| 2nd entry | 0.979 | 0.807 | 0.969 | 0.887 | 0.882 |
| 1st entry | 0.960 | 0.665 | 0.950 | 0.808 | 0.804 |
| s-TAS | 0.958 | 0.852 | **0.955** | 0.905 | 0.904 |
| | Fluo-N2DH-SIM+ | | | | |
| | DET | SEG | TRA | OP$_{CSB}$ | OP$_{CTB}$ |
| 3rd entry | 0.956 | 0.834 | 0.954 | 0.895 | 0.894 |
| 2nd entry | 0.981 | 0.813 | 0.973 | 0.890 | 0.889 |
| 1st entry | 0.984 | 0.682 | 0.957 | 0.809 | 0.804 |
| s-TAS | 0.972 | **0.822** | 0.971 | **0.897** | **0.896** |
| | PhC-C2DL-PSC | | | | |
| | DET | SEG | TRA | OP$_{CSB}$ | OP$_{CTB}$ |
| 3rd entry | 0.961 | 0.863 | 0.954 | **0.912** | **0.909** |
| 2nd entry | 0.983 | 0.821 | 0.975 | 0.896 | 0.895 |
| 1st entry | 0.967 | 0.715 | 0.959 | 0.841 | 0.836 |
| s-TAS | **0.972** | **0.720** | **0.966** | **0.846** | **0.843** |

Table 4: Detailed performance comparison of our approach with the top 3 performers on the leaderboard of the cell tracking challenge, as of 30[th] of January. The different color codes correspond to different teams, namely MU-Lux-CZ (as in Lux and Matula (2019)), ND-US, BGU-IL (as in Zhou et al. (2019)), CVUT-CZ, HD-Hau-GE, UVA-NL, HIT-CN, FR-Ro-GE, KTH-SE and TUG-AT.

