# OpenReview forum: "Siamese Tracking of Cell Behaviour Patterns"
_MIDL.io/2020/Conference — MIDL 2020_

### Official Review · AnonReviewer2 · 2020-02-25
**Good multi-stage solution**

**Rating:** 4
**Confidence:** 4

**Summary:**

This paper proposed a Tracking-Assisted Segmentation (TAS) method to leverage the segmentation performance on cells by adding siamese tracking information.
The performance is leveraged based on deep SiamFC tracker and U-Net segmentation.
Even each piece in the method is from the existing method, to aggregate the methods is not trivial.


**Strengths:**

The SiamFC tracker is pretrained on GOT-10k dataset to have better generalizability.
The method leverages the segmentation performance by separating the wrong segmentation cases such as collision and mitosis.
The comprehensive measurements and validations are presented, by comparing with the state-of-the-art methods and top performer in cell tracking challenge.
The values of using Collision detection and Mitosis detection are shown in Table 3.

**Weaknesses:**

The method is designed as a multi-stage system, which is not compared with the single-stage cell segmentation systems in MICCAI 2019.
We don't know the comprehensive comparison between TAS-general vs. TAS-specialised vs. Zhou et al from Table 2. Only Fluo-N2DH-SIM+ is provided for such comparison.
The writing and organization of the paper needs to be improved. Some contents are difficult to follow. For example, Mitosis detection section.
The performance is marginally worse/better compared with 1st place in the challenge.
The Figure 2 is not quite informative, maybe switch to a better visualization.

**Justification Of Rating:**

Even the system is designed as multi-stage. But we can see the author spent lots of effort to optimize the workflow.
The method is well validated with top teams in cell tracking challenge.
The  Collision detection and Mitosis detection are formed as detection problem to leverage the segmentation performance.

**Paper Type:**

methodological development

**Special Issue:**

yes

---

> ### Author Response · Authors · 2020-03-25
> **Rebuttal R2**
>
> Thank you very much for your comments and feedback. You have raised some important issues. Please, find our answers below.
>
> Single- vs Multi-stage approaches:
> We mostly focus on comparing with methods that competed in the ISBI cell segmentation challenge. The related work of Lux and Matula (2019) is a single-stage cell segmentation method and the work of Zhou et al (2019), which we compare our three TAS variants with, is a multi-stage approach. We will include further comparison with Perslev et al (2019) in future extensions of the current work. As they suggest, wherever multi-planar augmentations are possible, their model can produce state-of-the-art results, but for the 2D datasets of the ISBI cell tracking challenge, we expect their segmentation to be comparable with our TAS-general variant because of the same U-Net architecture.
>
> Table 1 reasoning for Fluo-N2DH-SIM+ dataset:
> The evaluations by the organizers are conducted one per month to avoid overfitting to the test set with multiple submissions. We chose Fluo-N2DH-SIM+ as a reasonable middle ground between the two more extreme datasets (DIC-C2DH-HeLa, PhC-C2DL-PSC). We have not evaluated the TAS-intermediate yet but will do so for the camera-ready.
>
>
> Unclear methodology, e.g., mitosis section:
> Thank you for pointing out the unclear text. We will add the following explanation to the mitosis subsection of the updated draft. $|M^i_{t-1}|$ indicates the number of cells that are associated (matched) in the subsequent frame, with cell $C^t_{t-1}$ in the previous frame. If the number of matched cells is zero ($M^i_{t-1} = 0$) then the cell has died; if the number of matched cells is one ($M^i_{t-1} = 1$) then there is only one cell in the next frame associated with the cell in the previous frame; if $M^i_{t-1} >1$ then there exists a mitotic event.
>
>
> Marginal differences:
> Indeed, at first sight we are marginally better. Upon further inspection, the 2nd and 3rd entries for all datasets in Table 4, all differences seem marginal, indicating saturation in the benchmark. We further explore this in our analysis of noisy labels with these properties: (1) Delayed mitotic events. (2) Pre-mitotic 2cell-single-cell labels (3) Cell death artefacts (4) Irregular shape of ground truth labels. We have experimented with the mitotic/collisional event timing by hastening and delaying the re-segmentation but noticed a decrease in performance. It is important to note that for high cell activity, such as in the PhC-C2DL-PSC dataset, we observe a bigger increase in performance due to our ability to correct the initial segmentation in difficult to observe cell collisions. Last, we emphasize that quantitatively speaking, we consider as much more important the fact that we produce state-of-the-art results on several different datasets and without any dataset-specific algorithms (at most, we only tune hyperparameters on the respective training sets); rather than the final numbers themselves. We incorporate these clarifications in Sec 4.
>
> Figure 2 vs Figure 3:
> Following your suggestions as well as a similar remark by AnonReviewer3, we swap Figures 2 and 3 in the main paper and the appendix. Thank you for your feedback.

---

### Official Review · AnonReviewer1 · 2020-03-12
**Siamese Tracking of Cell Behaviour Patterns**

**Rating:** 3
**Confidence:** 3
**Recommendation:** Oral

**Summary:**

The paper presents a method for segmentation and tracking biological cells in video sequences using Siamese tracking. The method includes an end-to-end cascade architecture to model biological cell tracking and predict collisions and mitosis. The evaluation is performed on three cell tracking challenge benchmark datasets.

**Strengths:**

The model is designed to be robust to morphological cell variations and predict events such as mitosis and collisions. The end-to-end cascade neural architecture and different configurations are explained for model interpretability. The validation is performed on three well-known benchmark datasets, where state-of-the-art performance is achieved for two datasets and second-best in the third dataset. The paper is well-written and organized.

**Weaknesses:**

The computational complexity of the proposed method and its comparison with state-of-the-art methods is not included. The authors have not emphasized on the clinical application of the method.  The manual tuning parameters in TAS-intermediate and TAS-specialised are not clearly explained. Details about experimental implementation and illustration are included in appendices, but these could be included in the main paper. The authors state in the contribution that the proposed method outperforms the state-of-the-art on three datasets but the results show this on two out of three datasets.



**Justification Of Rating:**

The paper describes a cell tracking method using Siamese tracking. The paper has a clear motivation and shows validation on three benchmark datasets with promising performance. The method demonstrates invariance to cell morphology and the ability to handle mitosis and collisions.

**Paper Type:**

both

**Special Issue:**

yes

---

> ### Author Response · Authors · 2020-03-25
> **Rebuttal R1**
>
> Thank you for your comments and feedback. We give our answers below.
>
> Computational complexity:
> Our model, U-Net and SiamFC combined, comprises 25 convolutional layers. In comparison, Lux and Matula (2019) use aU-Net with 20 convolutional layers. And, Zhou et al (2019) use U-Net Networks of 40 convolutional layers. Thus, our model sits between the two competitor methods. Generally, we needed around 5-10 minutes per sequence comprising 100 to 400 frames on a Geforce TitanX gpu.
> We add this clarification in appendix B.
>
> Clinical application:
> Our method can improve the autonomous segmentation of biological cells with the goal of inspecting multiple patient images in parallel, speed up diagnosis and ease the workload of doctors. Our approach indicates a more robust method of cell segmentation which will reduce the number of cells incorrectly detected and improve the accuracy and performance of such automatic systems. We update Sec 1 with these clarifications.
>
> TAS differences:
> As responded to AnonReviewer4, the three TAS variants differ in the pre- and post-processing steps.
> (1) TAS-general performs no pre-processing and no post-processing that relates to U-Net. That is, we (i) normalise the raw images to zero mean and unit variance and (ii) from the U-Net predictions we construct cell segments based on the euclidean distance, of the prediction, to the background. All parameters in TAS-general are the same across all cell types and datasets.
> (2) TAS-intermediate tunes the pre-processing hyperparameters (image normalisation, histogram equalisation and uneven illumination correction) independently per dataset (on the respective training sets) . This is to account for the very different range and distribution of pixel values that disturbs the U-Net. The siamese-based re-segmentation after the U-Net output (main contribution), remains the same for all datasets.
> (3) TAS-specialised tunes the pre- and post-processing hyperparameters per dataset, following Lux and Matula (2019). This variant is needed for detecting precise cell boundaries in order to minimise erroneous cell detection and be competitive with optimised state-of-the-art approaches. TAS-specialised retains the same re-segmentation augmented approach which we introduce.
> We add these clarifications in Sec. 2 under initial segmentation.
>
>
> Appendices in main paper:
> Following your suggestion and similar ones from reviewers AnonReviewer4 and AnonReviewer2, we move details on the methods and experiments in Sec. 2 and 3 of the main paper. For a precise list of the details added, please refer to the response to AnonReviewer4.
>
> Wrong statement of “outperforms the state-of-the-art on three datasets”:
> We are sorry for the misunderstanding. In the introduction we write “Our approach generalises well to different datasets outperforming state-of-the-art segmentation methods for biological cells on three benchmark datasets”, focusing on methods that are published in peer-reviewed conferences or journals. We replace the sentence with “Our approach generalises well to different datasets outperforming published state-of-the-art segmentation methods for biological cells on three benchmark datasets” and add a footnote for the third dataset.

---

### Official Review · AnonReviewer3 · 2020-03-12
**Siamese Tracking of Cell Behaviour Patterns**

**Rating:** 3
**Confidence:** 4
**Recommendation:** Poster

**Summary:**

The paper proposes a simple end-to-end cascade neural network to model the movement behaviour of biological cells and predict collision and mitosis events. They use U-Net for an initial segmentation and refine it further using a siamese tracker along the temporal domain. Their method demonstrates that this tracking approach achieves state-of-the-art results
on PhC-C2DL-PSC, Fluo-N2DH-SIM+ and DIC-C2DHHeLa dataset of the cell tracking challenge benchmarks.

The key ideas & experiments are well-written, explained and accompanied by source code.
The proposed Tracking-Assisted Segmentation (TAS) is achieved via - TAS General, TAS-Intermediate & TAS-specialized combined with Collision Detection, Mitosis Detection helps with re-segmentation and fine-tuning the final results.

The key significance of the presented ideas in this paper are:
Using siamese tracking for improved temporal correspondence and re-segmentation of erroneous predictions.
Robustness to morphology variations and able to model rare events such as mitosis, apoptosis and cell collisions.
Generalization on 3 different biological cell benchmark datasets & outperforming state-of-the-art segmentation methods.



**Strengths:**

The paper is well-written, with appropriate references, background research, benchmark datasets, sufficient experiments, experimental details and source code. The paper addresses the issue of cell tracking / segmentation while cells deform due to the process of mitosis and collision. They propose a Siamese tracking approach to detect such events and combine them with deep learning (UNet) and traditional computer vision (watershed) methods to achieve state of the art results.


**Weaknesses:**

It would be interesting to see some ideas from recent literature like “Transformers” / “Attention is all you need” and how could these be applied to cell tracking challenge.
Figure 2 in main paper is not that informative, in fact Figure 3 from the appendix Schematic representation of cell collision detection using a Siamese tracker which is a major contribution of this work.


**Justification Of Rating:**

The paper is well-written, with appropriate references, background research, benchmark datasets, sufficient experiments, experimental details and source code. The paper addresses the issue of cell tracking / segmentation while cells deform due to the process of mitosis and collision. They propose a Siamese tracking approach to detect such events and combine them with deep learning (UNet) and traditional computer vision (watershed) methods to achieve state of the art results.

The proposed methods and improvements are very valuable but needs more experiments and clarifications to be validated.


**Paper Type:**

both

**Special Issue:**

no

---

> ### Author Response · Authors · 2020-03-25
> **Rebuttal R3**
>
> Thank you for your comments and your appreciation. Below, we provide answers to your inquiries.
>
> Regarding transformer/fully attentional networks:
> Thank you for the suggestion, indeed this is a very interesting extension for future work. One way to incorporate such networks would be to use the attention features as a way to justify the mitosis and collision detection from the network itself. By motivating the image parts that contribute to the collision detection we can make the model more transparent and explainable.
>
> Figure 2 vs Figure 3:
> Following your suggestion, as well as a similar remark by AnonReviewer2, we swap Figures 2 and 3 in the main paper and the appendix. Thank you for your feedback.
>
> More  experiments and validations:
> As responded to AnonReviewer2, the organizers of the benchmark allowed for only a sparse evaluation of our algorithms (once per month evaluation). Following yours and AnonReviewer2’s suggestion, in the camera-ready version we will also include results for all datasets.

---

### Official Review · AnonReviewer4 · 2020-03-13
**Improved siamese cell tracking by the definition of cell collision and mitosis**

**Rating:** 2
**Confidence:** 5
**Recommendation:** Poster

**Summary:**

The authors propose a method that performs firstly, a 2D segmentation (U-Net) and secondly, a tracking (SiamFC). The results of the tracking are analyzed to determine which ones belong to cell collisions or mitosis. The identification of any of these two patterns is used to refine the initial segmentation.

The proposed approach is tested in three different datasets from the Cell Tracking Challenge and the authors show to improve some of the best accuracy measures reported so far.



**Strengths:**

While none the segmentation or the tracking are novel methods, the authors introduce the mathematical definition of collision and mitosis into the workflow and manage to improve the performance of the methods, especially in the cases in which cell displacements and/or density are large. this approach is interesting due to the lack of a priori probabilities as other tracking algorithms and the definitions provided are easy to understand/explain.

Additionally, the authors validate the proposed technique in publicly available datasets, which allows an objective benchmarking. Finally, they provide a Python code that can be used to analyze new datasets.




**Weaknesses:**

There are some parts of the methodology that are not very well written, which makes it very hard to understand.
It is not clear at all what are the differences between the three TAS, how the data augmentation was performed or how was the output of the U-Net processed.
The current definition of collisions or mitosis works only with time steps of size one. Especially for collisions, this might be critical as cells can be together for more than one frame. How does the algorithm deal with this? Besides, I wonder what is the effect of false negatives in this part of the tracking.
While the results suggest that this approach improves the tracking accuracy measures, could it be possible to give some examples of mitosis and collisions? Is it possible to visualize the real power of this part of the method? The example posted in Figure 2 is not very representative as part of the collisions are in the edges of the image.

**Justification Of Rating:**

From a high level, the approach proposed by the authors lacks of novelty. However, the definition of cell behaviors results in a more accurate tracking, as expected and this could be also a way to improve this common task.
I would strongly recommend the authors to elaborate more in the description of the method. The details regarding pre and post-processing, or model finetuning should be adequately reported.

**Paper Type:**

methodological development

**Questions To Address In The Rebuttal:**

- The authors say "the model is manually fine-tuned to the image properties", how exactly?
- Equation (1), could you please verify it is correct? if so, could you please give a better explanation of what is the difference between mitosis or M^i_{t-1} = 1?
- Is there any specific reason why the results in Table 1 are given only for one of the three datasets?
- I would change the lookup table (color map) in Figure 1 to the common grayscale one. It can be confusing and suggest different microscopy modalities.
- Figure 2, please correct the colors of each label. I guess that each cell is uniquely labeled and in the segmentation, there are two cells with the same color.
- Please elaborate more on Appendix B.
- I would suggest ordering each of the appendices according to the order in which the main text refers to them.
- The authors claim to detect apoptosis events, but they do not define it.

**Special Issue:**

no

---

> ### Author Response · Authors · 2020-03-25
> **Rebuttal R4, cont. 2**
>
> “manually fine-tuned to the image properties”:
> manual tuned parameters refer mostly to the pre- and post-processing details from the work of Lux and Matula 2019 where different sets of parameter values are used per cell type.
> For the pre-processing that entails using hyperparameters for image normalisation, histogram equalisation and uneven illumination correction, independently per dataset.  For the post-processing manually defined ellipsoid kernels, pixel value thresholds and filling gaps are used. We refer to this process as “fine-tuned” because it is specifically tailored to every dataset and its raw image properties.
>
> $M^i_{t-1} = 1$ vs Mitosis:
> $|M^i_{t-1}|$ indicates the number of cells that are associated (matched) in the subsequent frame, with cell $C^t_{t-1}$ in the previous frame. If the number of matched cells is zero ($M^i_{t-1} = 0$) then the cell has died; if the number of matched cells is one ($M^i_{t-1} = 1$) then there is only one cell in the next frame associated with the cell in the previous frame; if $M^i_{t-1} >1$ then there exists a mitotic event.
>
> Table 1 results for Fluo-N2DH-SIM+ only:
> The evaluations by the organizers are conducted once per month to avoid overfitting to the test set with multiple submissions. We chose Fluo-N2DH-SIM+ as a reasonable middle ground between the two more extreme datasets (DIC-C2DH-HeLa, PhC-C2DL-PSC). We have not evaluated the TAS-intermediate yet as per the request of you and other reviewers will do so for the camera-ready.
>
> Figure 1: Grayscale vs plasma color maps:
> Upon visual inspection the plasma colormap worked best for all datasets and we use it for consistency. We will also include the grayscale color maps in appendix C, although this may create visualization difficulties due to the uneven light illumination as explained in Ulman et al (2014).
>
> Regarding appendices:
> We move all requested details, as described above, from the appendices to the main paper. We reserve the appendices for additional information and for extra visualizations. Thank you for pointing this out. We will reorder the appendices as requested.
>
> Apoptosis definition:
> We add the definition of apoptosis in Sect 2: “Apoptosis is the cell death, or the disappearance of the cell from the field of view, as described by Ulman et al (2014), and is described by equation (1) in the main paper.”

---

> > ### Comment · AnonReviewer4 · 2020-04-03
> > **Response to authors**
> >
> > I thank the authors for their big effort in answering all the questions. I agree with all the remaining comments and I hope they implement all the changes they have mentioned.

---

> > > ### Author Response · Authors · 2020-04-03
> > > **Response to response**
> > >
> > > Thank you very much. We are in the process of updating the manuscript. Let us know if there is something more that we can do, if you believe it will help the quality of our work.

---

> ### Author Response · Authors · 2020-03-25
> **Rebuttal R4, cont.**
>
> How is the U-Net output processed and elaboration on appendix B:
> Using variants TAS-general and TAS-intermediate the output of U-Net is processed the same for all datasets. First, centroids are constructed using the euclidean distance, of every pixel, to the background. Centroids are first defined by regions where the euclidean distance is above a threshold value of 0.5; where 0 indicates background and 1 the farthest foreground. This will produce small centroid regions and in some cases where cells are too small no centroids. To remedy this a threshold value of 0.1 is used to define bigger centroid regions which are combined with the smaller ones. If there exist overlapping centroids from the two sets, then smaller centroids are preferred to split up cells that are collided. This process is repeated another time for threshold value 0, to detect even the smallest cells exiting or entering the frame. To remove unwanted artefacts in the centroids set, regions below the third of the average centroid region size are removed. This results in distinct centroids for each cell in the frame. In order to define where the boundary of lightly collided cells is, the random walker algorithm is used, using as seeds the centroid regions calculated. The output of the random walker approach produces individual cells on a frame and the frame is then fed into the siamese tracker for re-segmentation of heavily convoluted cells and cell behaviour modelling.
> Using the TAS-specialised variant, the same (post-)processing is applied on the U-Net output as in the work of Lux and Matula 2019. Namely, cell centroids are constructed using manually defined ellipsoid kernels, pixel value thresholds and filling gaps in the cell bodies. Every dataset is processed with a different set of parameter values tailored to its cell type. Then the watershed algorithm is run using the original U-Net prediction and the constructed centroids to produce individual cells in the frame. This frame is then fed to the siamese tracker for re-segmentation.
> We add these clarifications in Appendix B.
>
> Step size and false negatives:
> We have experimented with other step sizes as well, but found that mitotic events are not consistent in step size. Hence we have considered the step size to be a hyperparameter and from our parameter search, step size of 1 performed better.
> Indeed, cells can be together for more than one frame, however, at the first collision or mitosis the re-segmentation will correctly split the cells, and from that frame onwards, the algorithm will track both of them; re-segmenting them again if they stay collided. In the event of a false negative, e.g. a mitotic event that is not detected, the algorithm will assume the cell to be one bigger cell. In the next frame, if the mitotic event becomes more clear, i.e. the algorithm detects two independent cells, then the algorithm will re-segment the bigger cells into two, one frame too late. We add these clarifications in Sec. 2.
>
>
> Examples of mitosis and collisions and what is the power:
> We cannot put figures in the openreview comments and so we resort to a textual motivation. We notice that we improve particularly well when boundaries are not clearly distinguishable. From Figure 4, the mitotic event in the DIC-C2DH-HeLa dataset has a very visible effect. Hence a simple segmentation can detect the mitotic event. In contrast, the collision example in Figure 4 for the Fluo-N2DH-SIM+ dataset, does not always result in clearly defined boundaries. As Figure 3 indicates, when single segmentation cannot split collided cells, ours checks the next and previous frames for meshed bodies and is able to split them. The real power of the method, and where we get the biggest numerical improvements, is when there exist high cell activity in frames, such as collisions, for smaller cells such as is the case for the PhC-C2DL-PSC dataset due to the convoluted appearance of the cell bodies. We want to emphasize here that, as noted by the Reviewer, this advantages are inherent to the algorithm, and the core algorithm does not need to change (at most only hyperparameters tuned depending on TAS).
> We add these clarifications in Sec. 4.
>
> Figure 2 vs Figure 3:
> Figure 3 provides an additional illustration of the re-segmentation procedure while Figure 2 was meant to compare the performance of the re-segmentation with no cell behaviour modelling methods and related work. Given your remark we swap Figures 2 and 3 to visualise our method and make a stronger argument. In addition, regarding the colour of the cells in Figure 2, the cells are of different colour but due to their difference being around 1 in pixel value and the maximum value is around 40 it does not show it nicely. We fix this by recolouring the cells to distinct and clearly visible colours.
> We update Figures 2 and 3 in the camera ready version accordingly.

---

> ### Author Response · Authors · 2020-03-25
> **Rebuttal R4**
>
> Thank you very much for your comments and feedback and for recognizing the generality of our method. You raise important questions, which we address next. We also revise and clarify the camera-ready version accordingly.
>
> Novelty:
> We introduce a re-segmentation approach, which relies on Siamese trackers, combined with the watershed deconvolution method, to track cells and model cell collisions, mitosis and apoptosis. Compared to the original siamese trackers, as in the work of Tao et al (2016) and Bertinetto et al (2016) proposed on natural image videos, we use the model cell behavioural patterns in order to correctly track cells. Compared to previous cell tracking methods, such as in the work of Magnusson and Klas (2016) and Lux and Matula (2019), we generalise tracking to a cell type independent approach predicting cell displacement across frames.
>
> TAS differences:
> In the following, pre-processing refers to thresholding and filtering the raw image, e.g., to account for varying illumination, cell size, morphology and so on. With post-processing, we refer to additional adjustments after the U-Net output, such as filling prediction gaps, threshold smaller regions and fitting ellipses. We follow Lux and Matula (2019) for the pre- and post-processing steps, either per dataset -as originally in their work- or across datasets. The three TAS variants differ in the pre- and post-processing steps.
> (1) TAS-general performs no pre-processing and no post-processing that relates to U-Net, specific to a cell type. That is, we (i) normalise the raw images to zero mean and unit variance (ii) construct cell centroids from the U-Net predictions based on the distance to the background (where 0 indicates background and 1 the farthest foreground) and (iii) define distinct cells by running the random walker algorithm. All parameters in TAS-general are the same across all datasets.
> (2) TAS-intermediate tunes the pre-processing hyperparameters (image normalisation, histogram equalisation and uneven illumination correction) independently per dataset (on the respective training sets) . This is to account for the very different range and distribution of pixel values that disturbs the U-Net. The siamese-based re-segmentation after the U-Net output (main contribution), remains the same for all data sets.
> (3) TAS-specialised tunes the pre- and post-processing hyperparameters per dataset, following Lux and Matula (2019). This variant is needed for detecting precise cell boundaries in order to minimise erroneous cell detection and be competitive with optimised state-of-the-art approaches. TAS-specialised retains the same re-segmentation augmented approach which we introduce.
> We add these clarifications in Sec. 2 under initial segmentation.
>
> Data augmentation:
> Based on Ulman et al (2014), we add (1) random additive noise, (2) pixel value range shifts, and (3) maximum value cutoffs. Specifically, for each image we create 15 augmented images, 5 images per augmentation technique: (1) We add to the input image random Gaussian noise $N(0, std)$, with five possible $std$ standard deviations {0.1, 0.325, 0.55,  0.775, 1}. (2) We add an offset to the input image pixel values, which is a random number drawn (per image) uniformly in the range of [-1, 1] . (3) We set the maximum or minimum value of an image to a cutoff value calculated as $cutoff = max * num$. $max$ is either the minimum or the maximum value of the image, chosen randomly. $num$ is a random number drawn from a uniform distribution in the range of [0, 1]. E.g.: An image with range [-3, 3] and $max = -3$ and $num = 0.5$, will result in an image of range [-1.5, 3] with any value smaller than -1.5 being set to -1.5.
> We add these clarifications in Appendix B.

---

### Meta-Review · Area_Chair1 · 2020-04-02
**MetaReview of Paper109 by AreaChair1**

**Rating:** 3
**Recommendation For Accepted Papers:** Poster

**Metareview:**

Reviewers are in overall favor of this work. Despite some concerns about the individual processing stages not being overly original, their composing into a working pipeline seems to be non-trivial. Accompanied by the extensive experimental evaluation, there is good evidence that this contribution is worthwhile to be presented at MIDL. Also, authors went into great detail in explaining and reacting to critical reviewer comments regarding structuring and better explanations in parts of the work, which leads me to expect that the final paper has potential for a good MIDL contribution.

**Paper Type:**

both

**Special Issue:**

no

---

### Decision · Program_Chairs · 2020-04-11

Accept